# In their own words: A qualitative examination of student experiences with high-impact practices during the second-year transition

Austin L. Zuckerman[1,2,3¤], Gregory J. Stocker[1], Cheyenne N. Mercer[1], Randy G. Tsai[1], Thomas J. Bussey[3,4], Stanley M. Lo[1,3,5]*

1 Department of Cell and Developmental Biology, School of Biological Sciences, University of California San Diego, La Jolla, California, United States of America, 2 Joint Doctoral Program in Mathematics and Science Education, San Diego State University, San Diego, California, United States of America, 3 Joint Doctoral Program in Mathematics and Science Education, University of California San Diego, La Jolla, California, United States of America, 4 Department of Chemistry and Biochemistry, University of California San Diego, La Jolla, California, United States of America, 5 Research Ethics Program, University of California San Diego, La Jolla, California, United States of America

¤ Current Address: Department of Ecology and Evolutionary Biology, Cornell University, Ithaca, NY, United States.

* smlo@ucsd.edu

## Abstract

Researchers and practitioners have called for the use of high-impact practices to support student engagement and development in higher education institutions in the United States. Many studies have used quantitative methods to validate the importance of these practices in supporting broad academic and social outcomes, but fewer have used qualitative approaches to understand the range of outcomes that students perceive they are obtaining from these experiences. The development and evaluation of high-impact practices cannot be fully realized without leveraging student voices to understand the range of potential benefits that students acquire. Identifying practices that students perceive as valuable to their learning is essential for cultivating meaningful experiences that support student development and improve affective dispositions toward educational experiences. Second-year students are a particularly understudied population in higher education, facing unique challenges such as the "sophomore slump" that warrant increased access to high-impact practices. To complement existing literature on high-impact practices and second-year student development, this study applied a phenomenographic approach to analyze students' experiences in a summer-bridge program that supported students (n = 133) through the second-year transition. Using weekly written reflections as a primary data source, student experiences and outcomes were examined across four dimensions of student development: academic, social, professional, and personal. Students reported a variety of positive outcomes from their consistent participation in these practices, with a range of benefits observed primarily in their academic, personal, and

**Data availability statement:** All relevant data for this study are publicly available from the OSF repository (https://osf.io/8p3yu/?view_only=c2252a84b94e47a2944adeb306dc11ae).

**Funding:** The summer-bridge program in this study was supported by the University of California San Diego. The funders had no role in study design, data collection and analysis, decision to publish, or preparation of the manuscript.

**Competing interests:** The authors have declared that no competing interests exist.

social enrichment experiences. Perceptions of professional development outcomes were notably less salient and less detailed compared to the other three dimensions, suggesting that the types of activities students chose in this category may have offered fewer immediate benefits. Implications for cultivating meaningful experiences in higher education that can support second-year students' transition and development are discussed.

## Introduction

Higher education continues to be a pathway for social mobility and economic opportunities for an increasingly diverse United States population. The unemployment rate for graduates with bachelor's degrees is about half of the national average [1], and graduates with higher levels of education earn higher salaries and enhance the broader economic and social mobility of their communities [2]. Higher education researchers and practitioners continue to support the active recruitment of more diverse populations of students into higher education to improve educational and economic opportunities and outcomes for all students [3,4].

Research on student experiences and outcomes in higher education has been primarily focused at the first-year level [5,6]. High first-year attrition rates reflect the academic and social challenges that students face during their initial transition into the university, and higher education researchers have implemented various strategies and interventions to improve student retention at this stage [7–9]. Over the past decade, first-to-second-year attrition rates average approximately 23% despite modest increases in retention following the COVID-19 pandemic [10]. An additional 10% of students depart following their second year, with attrition rates plateauing in subsequent academic years [11]. Although attrition rates are highest in the first year, this continued attrition in the second year reflects unique challenges that students face beyond transitioning to the university [12]. However, there is significantly less research on undergraduate students' experiences during later stages, especially the second-year transition.

Existing literature on second-year student development and retention has primarily used quantitative methods to examine the efficacy of high-impact practices, which represent a variety of active learning practices and educational experiences that support students' unique needs and development [13–16]. High-impact practices aim to improve student outcomes through enrichment activities such as learning communities and service-learning experiences that cultivate global and local skills, knowledge, and competencies [17–19]. These studies have suggested the importance of programmatic efforts to facilitate practices that increase student involvement and engagement. However, there is a dearth of literature that examines the value of these practices and experiences from the students' perspectives, and few structured programs are targeted toward encouraging students' consistent participation in these practices as they enter their second year. Empowering students with agency and voice in these practices is essential for holistically assessing the overall quality

of these practices in supporting student engagement and improving students' affective dispositions toward educational experiences [20].

A qualitative approach was chosen for this current study to explore nuanced yet context-dependent experiences that are not readily accessible through standardized quantitative instruments. Programmatic interventions that include a qualitative examination of enrichment experiences from the student perspective are needed to determine the specific outcomes that second-year students perceive they are obtaining from these experiences and what individual practices and experiences support different dimensions of their development [21]. Having students explain their experiences in their own words can be used to identify practices that they buy-in to as enjoyable, valuable to their learning, and meaningful for multiple dimensions of their development [22,23]. To complement existing quantitative studies on second-year student experiences and development, this study asked: What experiences and outcomes do rising second-year students perceive they are obtaining from a novel summer-bridge program that facilitates high-impact practices to support their academic, social, professional, and personal development?

## Literature review

### Challenges and interventions during the second year

The second-year undergraduate experience is given considerably less attention in the literature, yet rising second-year students face unique challenges [24]. Second-year students are challenged to finalize a choice of major, complete more challenging coursework, and face greater autonomy and independence when making important academic and career decisions compared to first-year students [25–27]. Students in this academic stage often experience a phenomenon called the "sophomore slump," which is a period characterized by decreased satisfaction with the collegiate experience due to poor academic performance during the first year and other challenges related to their academic and social development [28,29].

Existing interventions for second-year students have included retreats, advising, seminars and courses, and residence hall programming [6,13,30–32]. Studies on interventions at the second-year level have also broadly focused on developing learning communities that aim to foster academic and social growth [21,33]. However, higher education researchers and practitioners also recognize the importance of outcomes related to students' development of cognitive skills and intellectual dispositions, occupational attainment, and preparation for adulthood and citizenship [34]. Student perceptions of utility value in enrichment activities have been suggested to be important for increasing intrinsic motivation, effort, and performance, with strong perceptions of utility in one activity having the potential to confer positive outcomes or implications in other contexts [35–37]. While several models focus directly on the practices and challenges of implementation, there remains a gap in understanding the range of outcomes that students perceive they are experiencing in these activities. Here, we review four salient dimensions of student experiences that have been identified and targeted for development: academic, social, professional, and personal.

### Academic development

Inequitable academic preparation resulting from differential access to resources can have enduring effects on academic performance leading into the second year [38]. Academic support programs that include tutoring and supplemental instruction are critical for academically underprepared students or students who struggle with the transition to more rigorous academic coursework in the second year [39,40]. Students rely on clear and consistent expectations, and the ability to meet these expectations is dependent on the level of academic support that they consider available and accessible to them. Student-faculty interactions have also been connected to increased learning outcomes and feelings of institutional connection [41–43]. Receiving feedback and support within the institutional community is critical for students to cultivate a sense of academic validation and belief in their abilities to succeed.

## Social development

Social support arises from shared activities within a group, community, or cohort, and positive social experiences have been shown to improve students' sense of belonging and connection to their institutional communities [43]. Cultivation of these networks depends on access to sufficient social capital [44]; however, without institutional support that helps them build this capital, students may experience a sense of isolation that undermines their social identity and sense of belonging. Poor social experiences and outcomes could also negatively impact students' perceptions of their campus climate, resulting in an emotional and cognitive load that can decrease academic motivation and performance [18,45]. For students who do not have access to social and cultural capital in navigating through their institutions, support in the form of peer mentoring and faculty advising can be significant determinants of whether students choose to continue their education [46,47]. Peer-led social support programs have successfully promoted positive social outcomes, as students can connect and share experiences with peers facing similar challenges in adjusting to their institutional environment [48].

## Professional development

The second year is when students increasingly engage in self-reflection and exploration around their future career pathways [49]. Because second-year students are often required to finalize their choice of major and draft tentative career plans, they are challenged to reflect on their strengths, interests, and values as they imagine their goals for life after their undergraduate education [25,50]. In addition to academic and social support, researchers and practitioners have called for increased implementation of high-impact practices that support career development and exploration earlier in students' educational trajectories [51]. These practices center on identifying students' professional values and increasing their occupational engagement in a range of career and vocational enrichment activities [51–53]. High-impact practices that engage students in these activities have been primarily integrated into academic curricula and course seminars [54,55]. However, workshops on career development topics, faculty panels, and career advising are other practices and services that have been recognized as potential agents of positive occupational development [56].

## Personal development

Finally, personal independence and self-efficacy are documented outcomes in programs and practices that integrate community service and service-learning [57]. High-impact practices that target academic, social, and career development can also lead to an increased sense of personal responsibility for one's learning and development. However, by engaging in meaningful activities beyond a purely academic setting, students also cultivate lifelong learning skills that are assimilated in other meaningful ways to prepare for adulthood and citizenship [58]. A broad range of activities can support these outcomes and can overlap with the other three dimensions of student development.

## Methods

### Study site and participating students

Data for this study were collected from a summer-bridge program that supported rising second-year students in the summer after their first year. The program was offered at a public, large, primarily residential, R1 doctoral university in the southwestern United States [59]. While an R1 institution was a data collection site, undergraduate research training was not a focus of this program. The program was modeled after existing institutional programs and policies that implement high-impact practices to improve students' satisfaction with their campus environments and collegiate experience [38,41]. Program activities included weekly social events for all participants, access to supplemental instruction through partnerships with the university learning commons, and workshops offered by the learning commons and career center. All students were assigned to peer mentors in small cohorts who facilitated social events and provided individualized

support. The program also hosted occasional professional development events, such as a faculty-student mixer. Students frequently reflected upon their participation in these activities in their weekly reflections.

To facilitate a residential learning community on campus and ensure that all program opportunities and resources were accessible, all participating students were provided with a stipend to cover on-campus housing expenses. The program spanned two five-week summer sessions, and students were required to complete one course during each session. Students who began their undergraduate studies in developmental writing and/or mathematics courses were selected on an invitation-only basis, and primary eligibility was contingent on good academic standing during each quarter of the first year and pace of academic progress (i.e., enrolling in fewer than the requisite number of credits per term to be on track for a four-year graduation), rather than on academic performance (e.g., course grades).

With the ease of obtaining a high sample of responses from the implementation of mandatory weekly reflections, choosing such a summer-bridge program was a convenient sample to catalog a broad range of student experiences. However, this sample of second-years was also purposefully selected through criterion sampling based on two criteria [60]. First, this program was a structured environment that encouraged student participation in high-impact practices across the four targeted dimensions of student development. Second-year students who did not participate in this program may not have had the opportunity to pursue all these practices and thus would not allow us to compare the range of outcomes obtained from different experiences in each of the four dimensions. Second, a substantial proportion of students were from minoritized backgrounds (first-generation college students and under-represented minorities) and represented a broad range of majors and disciplines (Table 1). The students' diverse backgrounds may have provided a more comprehensive range of perspectives on the outcomes and usefulness of these experiences in supporting their development.

## Phenomenographic approach

This study applies phenomenography, a qualitative approach that examines the different ways people experience and understand a phenomenon [62,63]. We use this approach to identify and describe the range of student experiences in the program, with student engagement in high-impact practices as the primary phenomenon of interest (Fig 1). Phenomenography is particularly suitable for this study because students participating in similar activities may foreground and interpret aspects of their experiences differently. Within the phenomenographic framework, we draw on variation theory to describe how students participate in high-impact practices and catalog the range of outcomes that students perceive from these experiences [64].

Variation theory triangulates three different perspectives to characterize the learning phenomenon under investigation: the intended object of learning, the enacted object of learning, and the lived object of learning. In the context of our study, the intended object is the facilitation of high-impact practices as supported by previous literature; the enacted object is the activities and experiences students engaged in; and the lived object is the outcomes that students perceived they obtained from participating in these experiences (Fig 2). Triangulating these three perspectives is essential for identifying differences between hypothetical and actual outcomes that students perceive from their participation, which can inform strategies for facilitating high-impact practices that support student buy-in and engagement with these experiences.

## Data collection and analysis

To catalog a range of experiences and outcomes from participation in this program, weekly reflections written by the students were collected. These reflections required participating students to describe and justify their participation in activities within four broad categories corresponding to the four targeted dimensions of student development: academic coursework (academic), community engagement (social), career skills and readiness (professional), and general learning skills and practices (personal) (Fig 3). In the inaugural iteration of the program, students were required to justify the fulfillment of at least three of the four categories of their choice weekly. In the second iteration, students were required to complete one activity related to academic

**Table 1. Demographics and background of student participants in the program (n = 133 across. two cohorts).** Under-represented minority (URM) status refers to students who fully or partially identify as non-White and/or non-Asian. The term URM refers to racial and ethnic groups that are not proportionally represented within specific academic fields in comparison to the population as a whole in the United States. We acknowledge that such gaps in representation are consequences of education debt owed to students from minoritized communities, resulting from a combination of historical, economic, sociopolitical, and moral factors, leading to differential access to educational opportunities over time and ultimately observable differences in academic achievement and representation [61]. Nonetheless, examining representation provides a defined measure for quantifying disparities between groups.

| Demographics | Percentage |
| --- | --- |
| **Gender** | |
| Woman | 58% |
| Man | 42% |
| **URM Status** | |
| URM | 66% |
| Non-URM | 34% |
| **College generation status** | |
| First generation | 64% |
| Continuing generation | 36% |
| **Discipline of study** | |
| Biological Sciences | 29% |
| Social Sciences | 20% |
| Engineering/Computer Science | 14% |
| Physical Sciences | 11% |
| Cognitive Sciences | 7% |
| Undeclared | 6% |
| Mathematics | 4% |
| Data Science | 3% |
| Environmental Sciences | 3% |
| Arts and Humanities | 2% |
| Health Sciences | 2% |

coursework and one of the other three categories. The program gave the participating students full autonomy to select any relevant experience or activity that they perceived would broadly develop their skills in each of these categories.

For each submitted reflection, students were prompted to identify an activity that fulfilled the category and what they learned from their participation in that category's activities. In the first iteration of the program, students submitted their responses without a structured template. In the second iteration, students were provided with a template that prompted them to (1) describe the activity and (2) describe what they learned from their participation in the activity. The students' responses were checked periodically by their peer mentors for completion to ensure students were engaging in appropriate program activities. These administrative checks did not include substantive feedback that would have influenced the content of students' reflections, thereby reducing potential bias in students' self-reported experiences.

Two iterations of the summer-bridge program had previously been completed, and student reflections were first accessed for the purpose of data analysis in this study on January 15, 2023, more than four years after students had completed the program. Participant consent was not required, as analysis of student reflections did not contain identifiable information and was considered quality improvement for evidence-based practices. The Institutional Review Board (IRB) under the Human Research Protection Program (HRPP) at the University of California San Diego (UCSD) provided a Not Human Subjects Research (NHSR)

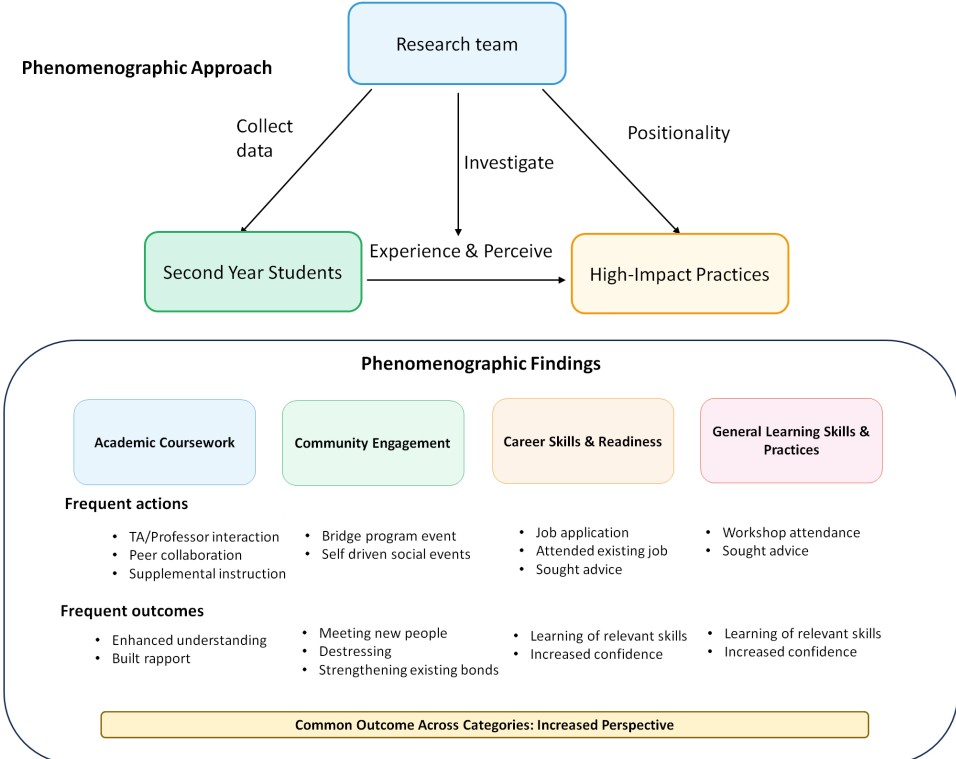

**Fig 1. Phenomenographic analysis of second-year students' experiences with high-impact practices.** Phenomenographic findings are summarized across four categories that align with the four dimensions of student development outlined in the Literature Review: Academic coursework (academic), community Engagement (social), career skills & readiness (professional), and general learning skills & practices (personal)..

determination for submitted protocol #802714: The above referenced project has been reviewed by the Director of the UCSD HRPP, IRB Chair, or IRB Chair's designee and is certified as not human subject research according to the Code of Federal Regulations, Title 45, part 46 and UCSD Standard Operating Policies and Procedures; and therefore, does not require IRB review.

For the preliminary analysis, inductive codes were written and applied to individual student reflections following techniques outlined in Saldaña (2021) [65]. The reflections were coded on an action-outcome axis. The action describes the general activity that students participated in to justify fulfillment of the program requirement. The outcome describes what the student perceived that they learned or gained from participating in that activity. The coding process began after all reflections from the first program iteration were collected. Memos and notes were generated after sampling through the first two weeks of reflections for each category. Reflections were annotated with brief labels, and emerging codes were frequently compared against earlier codes before a set of codes was finalized. These codes were used to summarize and describe salient insights from the raw data, and definitions were modified and expanded as more reflections were analyzed. The codes were applied to the complete data set after the coding scheme for each category was finalized.

## Reliability and trustworthiness

For initial training on coding, two researchers independently wrote preliminary codes for actions and outcomes for one of the four categories. After several iterations of discussing and revising the preliminary codes, the two researchers

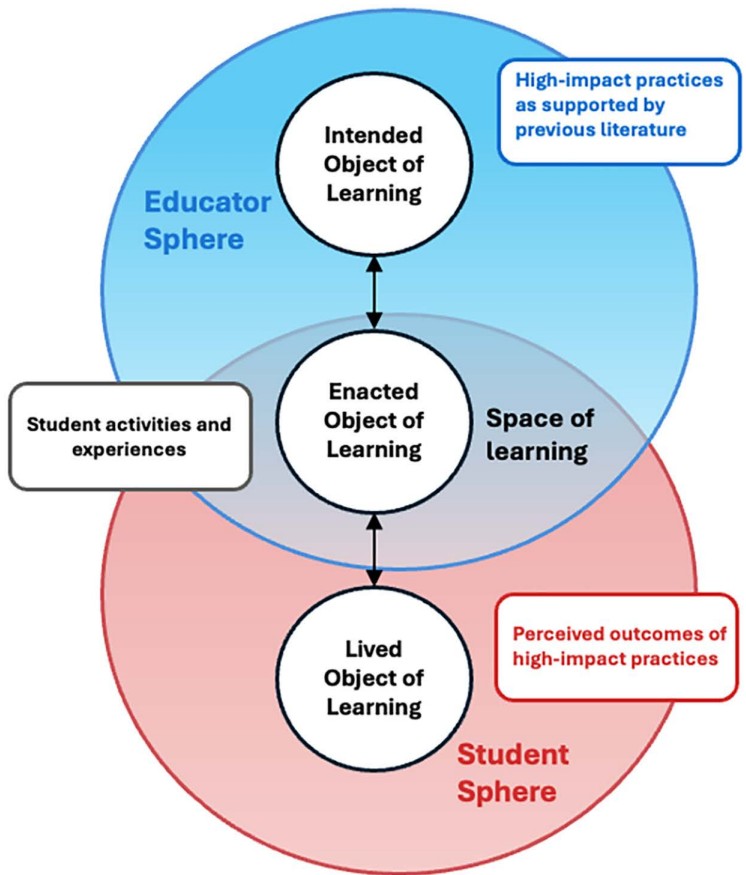

**Fig 2. Variation theory framework applied to second year student experiences with high-impact practices.** Three objects of learning within overlapping educator and student spheres: the intended object (literature-supported HIP goals), the enacted object within the space of learning (actual student activities and experiences), and the lived object (perceived outcomes).

independently applied the codes to individual student reflections. All disagreements were resolved through dialogic discussions to reach consensus. With a raw inter-rater agreement of 91%, one researcher proceeded with writing and applying codes to all reflections for the other three categories. Because of slight variations in the structure of the reflection prompts between the two iterations of the program, a similar training and analysis protocol was used on data collected in the second iteration. At least 10% of reflections in each category were coded by a second researcher to verify inter-rater agreement. An overall raw inter-rater agreement of 92% and a Cohen's kappa of 0.75 confirmed a substantial agreement between coders across all analyzed reflections (Table 2) [66,67].

Code and meaning saturation in this study were achieved by collecting data over multiple periods (across two iterations of the summer programs with different groups of students) and collecting and analyzing the data through the constant comparative method [68]. Patterns in the data set were summarized into codes using data collected in the first iteration, and code definitions were validated and refined by data from the second iteration. Some new codes emerged in the data from the second iteration of the program, and all codes used to analyze reflections from the first program iteration applied to reflections from the second program iteration.

While the IRB determined this was not human subjects research, several ethical considerations were addressed in the handling of student reflections. All student names and identifying information were removed from reflections before

# Weekly Reflection Instructions

Title: Week #____ Reflection

Scholar's Name: First Name Last Name

Date of Submission: MM/DD/YY

Describe and justify your weekly activities below. This question will always be some version of "tell me what you did this week and explain why it should count towards the [program]." Again, there are no right or wrong answers to this question; however, your Peer Mentor will be looking to see if you have effectively communicated what you did and why it should be counted towards the program weekly expectations. On average, you should be completing a minimum of three activities in three different categories per week. Consider the same guiding principles when you are writing: be honest, be personal, and be specific.

a. **Academic Coursework:**
   i. These are activities directly related to your enrolled coursework that goes beyond showing up to class and completing required assignments, assessments, and activities.

   ii. Explain what you did this week and why it should be counted in this category.

b. **General Learning Skills and Practices:**
   i. These are activities that more broadly develop your skills as life-long learners.

   ii. Explain what you did this week and why it should be counted in this category.

c. **Community Engagement:**
   i. These are activities that support the development of social and service-learning community networks.

   ii. Explain what you did this week and why it should be counted in this category.

d. **Career-Readiness Skills and Practices:**
   i. These are activities that develop and support transferable career or entrepreneurial skills.

   ii. Explain what you did this week and why it should be counted in this category.

**Fig 3. Sample instructions for weekly reflection on participation in program activities.**

analysis. The reflections were accessed only by authorized research personnel for program evaluation purposes. Trustworthiness and credibility of the research team were also considered in assessing the reliability of the interpretations. Data collection and analysis were conducted by a group of researchers from different sociocultural backgrounds and academic career stages, including undergraduate students, graduate students, and faculty. Student names and identifying information were removed from the reflections before being analyzed. The faculty were program coordinators who had interacted extensively with the participating students. The analysis was completed primarily by student researchers who had no prior interactions with the students or involvement with the program, allowing for a more objective interpretation of the students'

**Table 2. Interrater agreement and Cohen's kappa statistics for each coded category.**

| Category | Raw interrater agreement | Cohen's kappa |
|---|---|---|
| Academic Coursework | 93% | 0.77 |
| Community Engagement | 92% | 0.75 |
| Career Skills and Readiness | 88% | 0.52 |
| General Learning Skills and Practices | 92% | 0.60 |
| Overall [all reflections] | 92% | 0.75 |

experiences. Regular meetings between the research team members allowed for collaborative interpretations of the data and dialogic consensus to resolve disagreements. The student researchers were able to relate and compare their own diverse perspectives and experiences to those of the program participants. The faculty were able to then offer additional insights into the program structure that contextualized these experiences. Additionally, research findings were presented to various communities of education researchers at regional conferences, research showcases, and research meetings. Incorporating feedback from these sources verified that interpretations and claims were grounded in the data.

### AI usage

Claude.ai was used to assist with the conceptualization and layout of Figs 1 and 2. The outputs were evaluated by the authors for accuracy. Portions of the final figures were adapted from the AI-generated images but created independently by the authors.

## Results

The program participants' perspectives and experiences provide insights into the perceived range of outcomes obtained from a variety of enrichment activities, thereby empowering students with a voice in the quality of these activities. Code frequencies and definitions for activities that students used to justify fulfillment of the program category (action) and the perceived outcomes from that activity (outcomes) are summarized in Tables 3–6. The frequencies of the linkages between the actions and outcomes for each category are summarized in Fig 4.

### Academic coursework

Students were encouraged to participate in activities directly related to enrolled coursework that went beyond showing up to class and completing required assignments, assessments, and activities (Table 3, Fig 4A). The most common activity that fulfilled this requirement was a "Teaching Assistant (TA)/Professor Interaction." Many students perceived this activity as being beneficial to their academic development, as most who participated in this activity had articulated an increased understanding of the course material ("Enhanced Understanding") or perceived that they had gained valuable life and academic perspectives from conversations with instructional staff ("Enhanced Perspective"). Many students also articulated a personalized connection with instructional staff ("Built Rapport"). For example, one student articulated all three of these positive affective outcomes from one interaction with their professor:

> For this week's academic coursework, I attended my teacher's office hours. I was able to understand the material that originally did not make sense and I was able to know my teacher on a personal note. After the math homework, we discussed the importance of being educated and understanding towards other's differences, which allowed me to see my professor as a person making it easier to ask questions in the future.

Students also frequently studied with friends and classmates and used supplemental instruction resources to fulfill the program requirement. Similar to their interactions with TAs and professors, students perceived an increased understanding

**Table 3. Definitions of codes for actions (white) and outcomes (grey) for Academic Coursework activities. The total number of individual reflections where code was detected are indicated in parentheses.**

| Code Name | Definition |
|---|---|
| Attend Class (72) | Attended class or discussion sections |
| Independent Studying (82) | Studied or worked on academic projects alone |
| Peer Collaboration (377) | Received advice from a peer (not a program peer mentor) |
| Supplemental Instruction (217) | Attended supplemental instruction or tutoring session |
| TA/Professor Interaction (555) | Interacted with Teaching Assistant (TA) or professor |
| Built Rapport (55) | Developed stronger bond with faculty or other students |
| Enhanced Perspective (390) | Gained perspective on course material or a life-application insight |
| Enhanced Preparation (72) | Felt better prepared |
| Enhanced Productivity (131) | Felt more productive in accomplishing tasks |
| Enhanced Understanding (384) | Increased understanding of course material |
| Negative (26) | Perceived a discouraging interaction or experience |
| Personal Growth (19) | Felt a sense of personal growth |

**Table 4. Definitions of codes for actions (white) and outcomes (grey) for Community Engagement activities. The total number of individual reflections where code was detected are indicated in parentheses.**

| Code Name | Definition |
|---|---|
| Extracurricular (59) | Participated in extracurricular or volunteer activity |
| Mentor Event (113) | Participated in group activity hosted by peer mentor(s) |
| Other (7) | Participated in activities not applicable to other categories |
| Program Event (370) | Participated in a social activity organized by bridge program |
| RA Event (30) | Participated in event hosted by residential assistants |
| Social Self Driven (147) | Participated in social activity facilitated by peers or self |
| Workshop (11) | Attended a skills development workshop |
| Destress (113) | Reduced stress |
| New People (185) | Met new people; built or expanded social network |
| Other (26) | Obtained outcome not applicable to other categories |
| Outside Comfort Zone (43) | Grew personally and exceeded one's state of comfortability |
| Perspective: Academic (37) | Gained new perspective on academic development |
| Perspective: Professional (23) | Gained new perspective on professional development |
| Perspective: Social (109) | Gained new perspective on social relationships |
| Stronger Bonds (59) | Deepened existing personal or community relationships |

of course material as well as new and enriching perspectives relevant to their academic development. A large proportion of students who used peer study groups also perceived an overall increase in confidence and motivation that allowed them to feel more productive within their study environment. Students recognized the value of using their peer networks to support their academic enrichment, showing that a positive social environment mirrored an increased sense of academic belonging [43,68].

**Table 5. Definitions of codes for actions (white) and outcomes (grey) for Career Skills and Readiness activities. The total number of individual reflections where code was detected are indicated in parentheses.**

| Code Name | Definition |
|---|---|
| Advising (19) | Sought advice or feedback from advising center |
| Advice: Mentor (34) | Received advice from peer mentor, boss, or success coach |
| Advice: Peer (31) | Received advice from a peer (not a program peer mentor) |
| Advice: Professional (42) | Received advice or interacted with professional or professor |
| Attended Job (133) | Attended existing job, internship, or volunteer position |
| Job Application (226) | Participated in activities related to job application process |
| New Job/Function (63) | Attended new position or learned new skill in existing position |
| Miscellaneous (8) | Participated in activity that did not fit other codes |
| Other Career (23) | Participated in other career related activity (e.g., individual plan) |
| Social Enrichment (7) | Participated in a professional social event (e.g., club, social rally) |
| Workshop (63) | Attended a skills development workshop |
| Confidence (62) | Increased motivation, productivity, or confidence |
| Perspective (166) | Developed new perspective on goals, future, or career skills |
| Preparation (44) | Felt better prepared for future projects/career |
| Relevant Skills (160) | Gained applicable skills for future |

**Table 6. Definitions of codes for actions (white) and outcomes (grey) for General Learning Skills and Practices activities. The total number of individual reflections where code was detected are indicated in parentheses.**

| Code Name | Definition |
|---|---|
| Advice: Mentor (89) | Received advice from peer mentor, boss, or success coach |
| Advice: Peer (60) | Received advice or collaborated with peer |
| Advice: Professional (39) | Received advice or interacted with professional figure |
| Advising (71) | Sought advice or feedback from advising center |
| Extracurricular (19) | Participated in extracurricular or volunteer activity |
| Independent Planning (47) | Planned courses or schedule independently |
| Other: Academic (20) | Participated in academic activity (e.g., office hours) |
| Other: Adulthood (7) | Completed an adulthood responsibility (e.g., budgeting) |
| Other: Career (4) | Performed independent research on careers |
| Other: Miscellaneous (9) | Participated in an activity that did not fit other codes |
| Other: Social (7) | Participated in other social enrichment activity or event |
| Workshop (172) | Attended a skills development workshop |
| Confidence (58) | Increased motivation, productivity, or confidence |
| Perspective (155) | Developed new perspective on goals or future |
| Preparation (69) | Felt better prepared for future endeavor |
| Skills: Career (40) | Enhanced job application or career skills |
| Skills: Communication (21) | Enhanced communication skills |
| Skills: Life (44) | Enhanced skills that can be applied to lifelong endeavors |
| Skills: Test/Study (79) | Enhanced studying or test-taking skills |
| Skills: Time Management (80) | Enhanced time management skills |

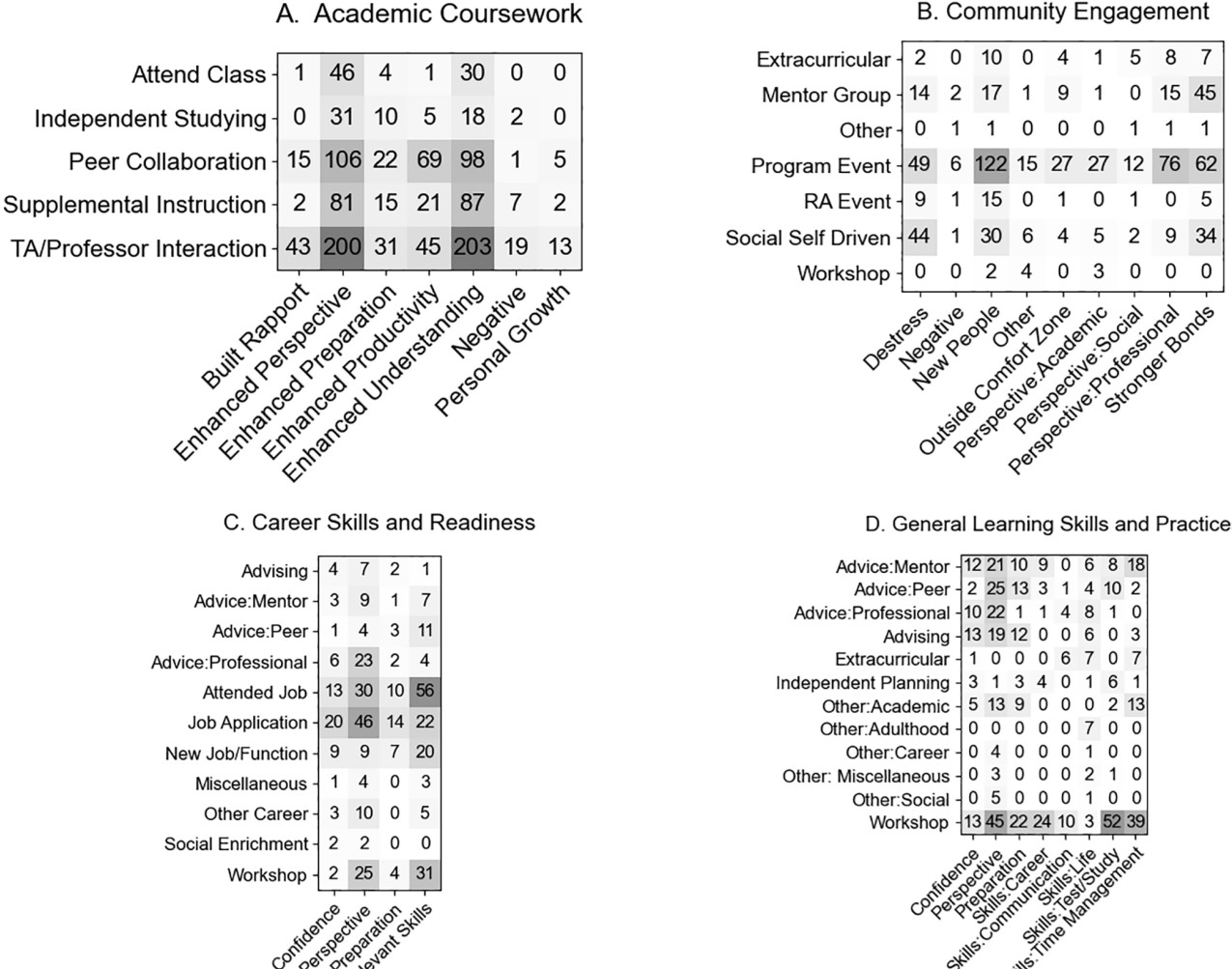

**Fig 4. Frequencies of the linkages between individual Action (vertical) and Outcome (horizontal) pairs. (A)** Academic Coursework, **(B)** Community Engagement, **(C)** Career Skills and Readiness, and **(D)** General Learning Skills and Practices.

Occasionally, students reported an interaction or experience that they did not perceive was beneficial for their learning, as indicated by the "Negative" code. These negative experiences usually reflected frustration when academic needs were not met, especially when students perceived that they were not given enough individual academic support when interacting with instructional staff. For example,

I attended office hours on Monday as my midterm was on Wednesday and I wanted to clear up some terms before the exam. It was a bit frustrating because my professor did not answer my question directly and he tends to go on random tangents regarding his own opinions or beliefs. I was however able to ask about an upcoming assignment which is an analytical essay. [...] Although my question was not answered I asked ahead for future assignments. The question that I asked in office hours was not as important as it did not appear on the midterm. Overall, I still felt like office hours are useful.

Overall, the frequency of negative outcomes was low. However, as this excerpt illustrates, students can still overall recognize the value of participating in supplemental academic enrichment activities even when experiencing frustrations or negative outcomes, highlighting the value of providing a supportive environment that encourages and values consistent participation in these activities.

### Community engagement

The program challenged students to participate in activities that supported social and service-learning networks (Table 4; Fig 4B). To fulfill this requirement, students most frequently attended activities that were facilitated directly by the program. The program facilitated a range of weekly structured social events, including team-building activities and informal social gatherings. Students perceived that these activities provided them with opportunities to meet new people and strengthen social relationships. With many opportunities to be involved in various social contexts within the program community, students also articulated that they had gained new perspectives or ideologies for improving their social development ("Perspective: Social"). For example,

> For the community requirement, I attended a [study jam]. While the focus was not on socializing, studying in the same room as my group/dorm mates truly humanizes them. I get a look at how they stress out, and how similar it is to myself. I feel [that] having the consideration to work quietly for the person next to you, is a way of showing love for the community. It shows a concern for the education of others.

Students were additionally placed into smaller cohorts under the leadership of a peer mentor. Unsurprisingly, the most common outcome described from participation in cohort activities was the strengthening of social bonds with the same peers over time. With access to larger program community events and smaller group activities, students were able to cultivate strong relationships while expanding their social networks in the program.

Students could also facilitate their own activities or participate in activities with individuals or other networks outside of the program community. Collectively, students described various positive affective outcomes from participating in these activities, including opportunities to destress, meet new people, and deepen existing relationships. For example, one student described multiple positive outcomes (coded as "Destress" and "Stronger Bonds") from a series of self-driven activities:

> My apartment mates and I went to the beach last Friday and bonded. We also took photos and watched the sunset. We also invited some other people from the program and watched movies on our laptops to de-stress from post midterms and quizzes and papers. I feel that the idea of coming together and bonding and de-stressing is something that is well needed. Being able to have company is healthy and a fun way to get to know others from this program.

The residential learning community formed during the program served as platform for students to form new friendships with other scholars, seek support networks with peer mentors, and draw upon new and preexisting social networks to cultivate their own meaningful social experiences. As rising second-year students experience a pivotal transition where they often must balance and compromise existing relationships while finding and navigating new networks that benefit their academic and career development, the program provided students with agency to leverage and maintain multiple social support networks inside and outside the immediate program community. Given that the perceived outcomes were similar in both structured program events and self-driven social activities, providing students with consistent access to structured social activities while also encouraging them to broaden their community engagement outside the program is optimal for maintaining and expanding meaningful connections as they prepare for the second year.

## Career skills and readiness

Students were challenged to participate in activities that developed and supported transferable career or entrepreneurial skills (Table 5; Fig 4C). The most common activities that students took to fulfill this objective were job searches or activities related to the job application process. Often, there were few detectable outcomes articulated from participation in this activity. Students often described new perspectives about the job application process, but these descriptions were typically brief or hypothetical. For example, one representative reflection stated: "[I] applied for multiple job positions through [the campus job portal]. I learned that the process can be often exhausting but with patience, I will be able to achieve a desired position."

Despite fewer instances, a broader scope of positive outcomes was described by students who attended new or existing jobs or internships. These students perceived that they acquired a range of relevant and transferable skills within a structured professional environment, including communication, networking, time management, and technical skills. Several students also articulated new perspectives relevant to their future career aspirations and increased feelings of motivation or preparation. For example, one student who held a tutoring position articulated:

> Going to work has help[ed] me exercise my public speaking skills and increase my confidence in being able to articulate complex course material for other people's understanding. I have also learned how much preparation goes into teaching material.

Other activities that were coded less frequently also provided students with enriching perspectives for their professional endeavors. For example, a student-faculty mixer (coded as "Professional Interaction") was offered as an opportunity for students to interact with various faculty and learn about their experiences. Several students received powerful perspectives that gave them new insights into their professional goals and identities. For example, one student articulated that a student-faculty mixer was a transformative experience that was particularly meaningful as a first-generation college student.

> This event was probably the best event I have ever attended at [the university]. It was so helpful, and so ensuring to know that it takes a long time to figure out your true passions in life. It really helped to see what to expect, especially as a first generation student. Often times, we need a lot more guidance by those who have simply succeeded in a profession, as there are not many people in our family who can give us that kind of advising.

Collectively, while many students opted to hone their career preparation by focusing on job and internship applications, they were able to describe more meaningful and tangible outcomes (e.g., the acquisition of relevant skills) when having opportunities to participate in other professional enrichment activities focused on networking and direct professional interactions (Fig 4C).

## General learning skills and practices

The activities in this category were aimed at having students broadly develop their skills as life-long learners (Table 6; Fig 4D). Students most commonly attended one of the "Skills Workshops" hosted weekly at the library commons. They articulated that these workshops helped them enhance a variety of skills, including time management, studying and test-taking strategies, and job application preparation. For the variety of activities that students pursued, the most common outcome described was perspectives for their objectives or future (coded as "Increased Perspective"). Often overlapping with activities that justified the "Career Readiness and Skills" requirement, the enhanced perspectives often resulted from meaningful conversations with mentor figures, including advisers, professors, and peers. For example, one student articulated that their professor's perspectives had increased their confidence in their personal development and career aspirations (coded as "Advice: Professional", "Confidence", and "Perspective").

This week I met with [my professor] who taught my lab course for chemistry. I think this meeting helped me gain insight on what I want in life. How much I should also expect from my major. The meeting gave me more inspiration to love math instead of run away from it and use this next course to get better at it instead of just passing it. I am happy to say I have more confidence after talking to her about careers and gained ideas on how to get into research programs.

Activities for this category often extended beyond the campus setting. Students explored activities that were not limited to their academic, social, or professional development, often pursuing activities that were associated with progression through early adulthood. Examples of these activities included cooking, learning how to drive, budgeting, and independent research on learning strategies. Students cultivated skills and perspectives useful for their life-long development (coded as "Other: Adulthood" and "Skills: Life"). For example, one student reported the following.

My suitemate and I planned a dinner event. In this event I was responsible to assigning the cooking, preparing tasks and I also had to go and buy the cooking material. This event taught me to be responsible since I had to supervise that everything runs smoothly and also improve my event planning skills.

While institutional intervention models typically focus on cultivating academic, social, and professional enrichment experiences, the summer-bridge program empowered students to pursue other opportunities and experiences that they deemed relevant to their lifelong learning. Although these experiences often overlapped with the other three dimensions, giving students leeway and voice to identify experiences that they perceived were beneficial to their general learning uncovered a broader range of experiences that students found meaningful to their development. Thus, having a flexible category where students could focus on other aspects of their personal development allowed the program to support their learning, engagement, and development more holistically than if they were constrained to only focus on activities related to the other three dimensions.

## Discussion

Rising second-year students often experience important academic, relational, and developmental transitions without institutional support, especially as institutional resources are directed primarily to incoming first-year students [69]. Previous research supports that positive affective outcomes related to academic and social engagement are strong predictors of student success and development [38,70,71], but second-year are provided with less programmatic experiences and resources that support these outcomes during a period of important academic and career decisions. There has been an increased call for interventions that target second-year students' needs and development to reinforce their institutional commitment, especially those that intentionally design the second year around high impact practices that connect students to developmentally purposeful activities. These activities, such as increased faculty connections and academic advising resources, have been shown to promote greater general success for second year students [15,69].

Situated in the existing literature, this study fills an important gap by documenting and analyzing outcomes of high-impact practices using a primarily qualitative approach to give students' voice and agency to reflect on the perceived usefulness of these experiences. While research continues to show that student participation in high-impact practices can predict student retention, previous recommendations for policy and practice have called for more comprehensive assessments of second-year experiences across institutional contexts to inform decision-making about implementation of meaningful and supportive practices [45,69]. The reflections on these experiences and outcomes described in this current study have implications for the development of practices and institutional strategies that can best support students in their transition to the second year.

First, the program challenged students to access a larger repertoire of academic resources that went beyond showing up to class and completing required assignments. The students perceived that these activities facilitated an increased

understanding of academic content, provided them with new perspectives on how to improve their academic development, and promoted positive social outcomes, such as increased rapport with professors and peers. For course instructors and support program coordinators, requiring or implementing incentives for students to participate in office hours, supplemental instruction, and peer study groups can be strategies that increase academic engagement and potentially improve academic outcomes.

Second, the program facilitated a variety of social events that supported the development of multiple social support networks as students were encouraged to expand connections and seek other social opportunities beyond structured program events. Students articulated social cohesion within the program community and gained broader perspectives about building relationships and developing strong support networks. Theories on social capital have established that students can gain a multitude of benefits from social networks, which include motivations to seek academic and institutional resources that support multiple dimensions of their engagement and development [72]. Because social capital can be mobilized into other forms of navigational resources, structured programming that requires second-year students to regularly participate in structured program events, networking opportunities with peer mentors, and independent social activities can provide students with multiple forms of personal and institutional networks that provide meaningful information and support.

Finally, because the second year is the academic stage where students often finalize their major and draft tentative career plans, students often need extra support and advice as they seek professional opportunities and internships within their fields. Career-related forms of mentorship typically involve a mentor that sponsors authentic assignments and practical experiences while coaching the student in their exploration of different career options [73]. Because students have diverse majors and career aspirations, facilitating these specialized opportunities for all students may be time-consuming and expensive at the program-level. Due to this limitation, students in this study most frequently utilized the job search and application process to fulfill the Career Readiness and Skills requirement, which yielded outcomes focused primarily on perseverance rather than meaningful career insights.

Unlike academic, social, and general learning activities that often provided immediate engagement and feedback, professional development activities that rely on external dependencies (such as the job application process) may not have outcomes that are immediately apparent to students. Additionally, brief programmatic duration may be insufficient for many students to develop meaningful professional relationships or complete substantive work experiences. If students cannot obtain or maintain a job or internship, they should be encouraged to pursue alternative enrichment activities that may provide more immediate learning outcomes. Increasing access to skill development workshops and faculty-student mixers may provide students with different professional perspectives, increased awareness of existing opportunities, and enhanced general career skills (e.g., communication and teamwork skills) that could be transferable to various professional settings.

## Limitations

There are several limitations to this study. First, not all reflections had a detectable outcome for each action described. Some participating students simply described the activities without elaborating on what they gained or learned. However, the aggregate sample size of reflections from both cohorts was substantial, allowing for a more comprehensive overview of perceived outcomes from different high-impact practices and experiences (Tables 3–6). Second, our study sample represents a small population of rising second-year students within a single university program. There may be variations in student experiences and perceived outcomes from these experiences across institutions with different student populations. Third, outcomes related to student retention (e.g., graduation rates) are not directly measurable in this study. This study examined different student experiences and the perceived outcomes and usefulness of these experiences, but the self-reported nature of the outcome data is not ideal for evaluating the overall efficacy of the program and measuring student success. Future longitudinal studies could focus on student outcomes during later stages of their educational

trajectories and examine metrics such as retention rates and major completion compared to matched controls who did not participate in the program. Qualitative follow-up studies at regular intervals could also be triangulated to assess sustained engagement in high-impact practices following the program, particularly the continued use of academic support services and peer and faculty networking. These studies would provide insights into whether short-term perceived outcomes translate into measurable long-term outcomes and assess the viability of programmatic efforts that support high-impact practices for improving educational and occupational attainment and institutional retention.

Student reflections in this study provided information-rich cases that represented a broad range of experiences and outcomes of those experiences. While we organized outcomes into four dimensions of student development (academic, social, professional, and personal), we recognize that these dimensions are not inseparable. For example, social support and belonging may reinforce academic self-efficacy, and career-focused activities can strengthen both academic engagement and social connections. Intersectional analyses could explore the extent to which these dimensions interact to inform how high-impact practices can support holistic student development. Additionally, reflections that prompt students to discuss other elements of their academic and social expectations and navigational processes in the university can elucidate a broader range of psychosocial experiences and challenges related to student identity navigation and development during the prospective second-year transition. Because the program supported students primarily from underserved backgrounds, investigating how different cultural dimensions of identity (e.g., first-generation status, race/ethnicity, gender, etc.) intersect with these psychosocial challenges can further inform practices and target interventions that are inclusive and supportive to diverse populations of rising second-year students. Beyond studies with general undergraduate populations, future research could also examine how high-impact practices support students across different academic and disciplinary contexts, including those with specialized training requirements [e.g., 74].

## Conclusion

This research contributes to a more comprehensive understanding of student buy-in to high-impact practices that support student development through the second-year transition. In accordance with other researchers and practitioners who have studied second-year student experiences, we recommend that colleges and universities allocate funding and resources to facilitate programs and interventions that make these experiences more accessible and structured for second-year students, who historically have been largely neglected as a target population for institutional support [75]. However, we also emphasize that the development and evaluation of these interventions cannot be fully realized without student voice in these experiences. We encourage campus administrative leaders to complete comprehensive audits of the second-year experience within their unique institutional contexts by encouraging direct student input through reflections, surveys, interviews, focus groups, and advisory panels with student representatives [45,69]. These efforts will illuminate the issues that rising second-year students feel are imminent in their educational trajectories and the desired resources and experiences that would allow them to thrive through this transition. Institutional interventions that strategically implement meaningful high-impact practices that target multiple dimensions of student development during the second-year transition can then evolve into more durable models that effectively counter the "sophomore slump" phenomenon in higher education in the United States.

## Acknowledgments

We thank the students participating in the summer-bridge program for sharing their perspectives and experiences.

## Author contributions

**Conceptualization:** Stanley Lo.

**Data curation:** Thomas J. Bussey.

**Formal analysis:** Austin L. Zuckerman, Gregory J. Stocker, Cheyenne N. Mercer, Randy G. Tsai.

**Funding acquisition:** Thomas J. Bussey, Stanley Lo.

**Investigation:** Stanley Lo.

**Methodology:** Austin L. Zuckerman, Stanley Lo.

**Project administration:** Stanley Lo.

**Supervision:** Thomas J. Bussey, Stanley Lo.

**Validation:** Austin L. Zuckerman, Gregory J. Stocker, Cheyenne N. Mercer, Randy G. Tsai.

**Writing – original draft:** Austin L. Zuckerman.

**Writing – review & editing:** Austin L. Zuckerman, Thomas J. Bussey, Stanley Lo.

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
