## [Decision Letter · Decision Letter 0]

28 Apr 2025

Dear Dr. Lo,

Thank you for submitting your manuscript to PLOS ONE. After careful consideration, we feel that it has merit but does not fully meet PLOS ONE’s publication criteria as it currently stands. Therefore, we invite you to submit a revised version of the manuscript that addresses the points raised during the review process.

We look forward to receiving your revised manuscript.

Kind regards,

Ahmed Abdelwahab Ibrahim El-Sayed, PhD, MSN in Nursing Administration, BSc in Nursing

Academic Editor

PLOS ONE

Journal Requirements:

For additional information about PLOS ONE ethical requirements for human subjects research, please refer to http://journals.plos.org/plosone/s/submission-guidelines#loc-human-subjects-research .

“The summer-bridge program in this study was supported by the University of California San Diego.”

“We thank the students from the summer-bridge program for sharing their perspectives and experiences. The summer-bridge program was supported by University of California San Diego.”

“The summer-bridge program in this study was supported by the University of California San Diego.”

5. In the online submission form, you indicated that “The datasets created and/or analyzed for this study are available from the corresponding author upon reasonable request.”

**Additional Editor Comments:**

Dear authors,

Thank for your study

Your contribution is original

Reviewers provided you with detailed advice to enhance your manuscript

Your manuscript has good potential for publication in PLOS ONE

best regards,

Reviewers' comments:

Reviewer's Responses to Questions

**Comments to the Author**

1. Is the manuscript technically sound, and do the data support the conclusions?

Reviewer #1: Yes

Reviewer #2: Yes

Reviewer #3: Yes

Reviewer #4: Yes

2. Has the statistical analysis been performed appropriately and rigorously?

Reviewer #1: N/A

Reviewer #2: N/A

Reviewer #3: N/A

Reviewer #4: Yes

3. Have the authors made all data underlying the findings in their manuscript fully available?

Reviewer #1: Yes

Reviewer #2: No

Reviewer #3: Yes

Reviewer #4: Yes

4. Is the manuscript presented in an intelligible fashion and written in standard English?

Reviewer #1: Yes

Reviewer #2: Yes

Reviewer #3: Yes

Reviewer #4: Yes

Reviewer #1: The article under consideration for publication is well-structured, addresses an interesting study problem, and makes a significant contribution to the design of educational practices relevant to the development of higher education students, particularly those in transition to the second year. The qualitative methodological approach employed by the authors, which involves giving students a voice, enables a more integrated or holistic understanding of how students perceive these practices and whether they are meaningful to them.

The title of the article is an accurate reflection of its content.

In the introduction, the authors present a detailed account of the background to the study and the research question, providing a comprehensive overview of the context in which it was conducted and its significance in terms of educational practice and policy. This framework enables the reader to comprehend the scientific and social significance of the study.

The literature review addresses the existing research on the problem and identifies the strengths and limitations of other studies in their approach to the topic. Despite the dearth of studies involving higher education students during the second-year transition, it would have been beneficial to incorporate some more recent literature into the analysis.

In the methodological section, the authors provide a comprehensive account of the processes and procedures employed to develop the study, delineate the methodological perspective adopted (phenomenology) and justify its selection, identify the method of data collection in detail, and furnish a detailed description of how the participants were recruited, including information on time span (dates) and procedures, with particular attention to those aimed at guaranteeing reliability in the analysis.

With regard to data analysis, the present study describes each step of the content analysis of the students' reflections, as well as the process by which the conclusions were generated. This includes the criteria used to generate the themes, the number of coders involved, and the means of ensuring consistency between evaluators. The manuscript addresses ethical issues in an appropriate manner.

The presentation of the results adheres to the themes that emerged from the analysis. It offers a reflective analysis of the data obtained, augmented with quotations from the participants that substantiate the themes and the authors' interpretations.

The discussion of the results synthesizes the data obtained and confronts it with the existing literature. The authors demonstrate a clear intent to translate the knowledge obtained into educational practice or policy, explaining the implications of their findings and presenting concrete proposals.

The study's limitations are presented and discussed in detail.

Recommendations:

It is recommended that the Literature Review Section include references from more recent publications. Furthermore, the references included in the references list/section should be reviewed and corrected.

Reviewer #2: This is a well-executed and timely study that adds meaningful qualitative insights into second-year undergraduate experiences, a population often overlooked in higher education research. The use of a phenomenographic framework is appropriate and thoughtfully applied. The authors do an excellent job of connecting their findings to existing literature and highlighting practical implications for institutional policy and student support.

However, to comply with PLOS ONE's data policy, the authors should ensure that anonymized reflection data or a suitable sample of coded data is made openly accessible through a repository or supplementary material or clearly justify any restrictions on sharing.

Although the manuscript does not include statistical analyses �as it is not required for the methodology� because this study was Qualitative, however, this should be clearly stated in the limitations to avoid confusion for readers expecting like quantitative measures.

Overall, this is a valuable contribution to the field and would benefit from minor revisions regarding data availability compliance.

Reviewer #3: Introduction:

High-impact practices (HIPs) are teaching and learning approaches that significantly impact student learning, engagement, and success. For second-year university students, HIPs can be particularly beneficial in fostering academic momentum, promoting student engagement, and enhancing career readiness.

I agree that high-impact practices offer a range of benefits for second-year university students, from improved academic performance to enhanced career readiness. By incorporating HIPs into curriculum design and providing support and resources, universities can promote student engagement, success, and career development.

However, it would be useful for readers to know whether some of the students had the opportunity to use research skills as part of the project because you mentioned R1 doctoral university as part of the data collection site.

Methods

Phenomenography requires the use of diagrams, tables or figures. Your readers will find a diagrammatic representation of your results helpful in the results section.

Ethics, study setting and participants

More information is required on the mandatory weekly reflections from the participants. You selected students based on their academic performance. It is interesting to note that ethical clearance deemed this project a quality improvement.

Conclusion

This study is vital to support an underserved body of university students. Thank you for undertaking the study.

Reviewer #4: Dear Editor

Thank you for the opportunity to review this manuscript. I appreciate the effort and dedication of the authors in conducting this important study. Below, I have provided my comments and suggestions, which I hope will contribute to further strengthening the manuscript.

Best regards,

Dr Ali Afshari

Length of the article:

The article is overly long at 8,352 words (including references), especially for a qualitative study. The literature review and discussion sections could be shortened to improve readability and focus.

Lack of clarity in some findings in the abstract:

The abstract does not fully identify key findings, such as the superiority of academic, personal, and social outcomes over professional outcomes. Also, there is no mention of the concept of “sophomore slump,” which is prominent in the main text.

Repetition across sections:

Some material in the literature review, findings, and discussion sections is repetitive. For example, descriptions of the four dimensions of student development (academic, social, professional, and personal) are overly repeated across sections.

Limitations of the study sample:

The study was conducted at only one university, and its sample (133 students) may not be representative of more diverse populations at other institutions. This limitation reduces the generalizability of the findings.

Lack of long-term outcomes:

The article did not examine long-term outcomes, such as graduation rates or student retention. This limits the scientific value of the study for evaluating the effectiveness of the program.

Inadequate explanation of consent:

Although it is noted that consent from participants was not required due to the lack of identifying information, more explanation could have been provided about the ethical considerations associated with using students’ personal reflections.

Limited results in the professional dimension:

The findings related to professional development are less prominent than other dimensions (academic, social, personal). The article could have analyzed this weakness in more depth or explained its possible reasons.

Repetition of some references:

The frequent use of a few specific sources (e.g. [31, 62]) indicates an excessive reliance on a limited number of studies. More diversity in citations could have strengthened the arguments.

Vague suggestions for future research:

The suggestion for longitudinal studies in the discussion section is general and does not provide specific details about the methods or expected results.

Complexity of some terms without adequate explanation:

Some technical terms, such as "variation theory" or "phenomenography," may not be entirely clear to non-specialist readers because sufficient explanations are not provided in the text.

**Do you want your identity to be public for this peer review?** For information about this choice, including consent withdrawal, please see our Privacy Policy

Reviewer #1: No

Reviewer #2: No

Reviewer #3: **Yes:** Adeniyi Olanrewaju Adeleye

Reviewer #4: No

---

## [Author Response · Author response to Decision Letter 1]

13 Oct 2025

Please see included cover letter for responses.

---

## [Decision Letter · Decision Letter 1]

20 Nov 2025

Dear Dr. Lo,

Thank you for submitting your manuscript to PLOS ONE. After careful consideration, we feel that it has merit but does not fully meet PLOS ONE’s publication criteria as it currently stands. Therefore, we invite you to submit a revised version of the manuscript that addresses the points raised during the review process.

We look forward to receiving your revised manuscript.

Kind regards,

Ahmed Abdelwahab Ibrahim El-Sayed

Academic Editor

PLOS ONE

Journal Requirements:

Additional Editor Comments:

Dear Authors,

Thank you for your revision.

One reviewer has identified several minor issues that still need to be carefully addressed before we can proceed to a final editorial decision regarding your manuscript. Please review his comments thoroughly and make the necessary revisions.

Reviewers' comments:

Reviewer's Responses to Questions

**Comments to the Author**

Reviewer #1: All comments have been addressed

Reviewer #3: All comments have been addressed

Reviewer #4: All comments have been addressed

2. Is the manuscript technically sound, and do the data support the conclusions?

Reviewer #1: Yes

Reviewer #3: Yes

Reviewer #4: Yes

3. Has the statistical analysis been performed appropriately and rigorously?

Reviewer #1: N/A

Reviewer #3: N/A

Reviewer #4: N/A

4. Have the authors made all data underlying the findings in their manuscript fully available?

Reviewer #1: Yes

Reviewer #3: Yes

Reviewer #4: Yes

5. Is the manuscript presented in an intelligible fashion and written in standard English?

Reviewer #1: Yes

Reviewer #3: Yes

Reviewer #4: Yes

Reviewer #1: All proposed amendments have been integrated into the manuscript, resulting in a revised and enhanced version. With these modifications in place, I believe the article is now ready for publication.

Reviewer #3: Thank you for taking the time and effort to respond to my suggestions. I hope your work contributes to the broader learning and teaching.

Reviewer #4: Dear Authors,

Thank you for submitting this insightful qualitative study. The revised manuscript is well-structured overall, with clear implications for higher education programs, and the use of weekly reflections as data sources adds authenticity to the findings. The issues addressed in the new version have strengthened the article. However, while the study makes a valuable contribution, I have some suggestions to enhance clarity, conciseness, and generalizability.

Abstract:

The abstract effectively highlights the gap in qualitative research on high-impact practices (HIPs) for second-year students, providing a compelling rationale for the phenomenographic approach. To enhance readability, consider briefly specifying one example of a 'positive outcome' in each dimension to give readers a more vivid preview of the results."

Elaborate slightly on how the summer-bridge program's design influenced these outcomes (e.g., activity types), as this could strengthen the link to practical recommendations without exceeding word limits."

Introduction

Condense the discussion of first-year comparisons (lines 88-94) slightly to allocate more space for emerging trends in second-year retention, such as post-pandemic effects, to underscore timeliness."

For greater impact, suggest explicitly noting potential overlaps between dimensions (e.g., how social support influences personal efficacy, as hinted in lines 191-192) to invite future intersectional analyses."

To extend applicability, I recommend incorporating a brief discussion of how these challenges manifest in nursing and paramedical students, who often grapple with unique educational hurdles like integrating clinical simulations, managing high-stakes ethical dilemmas, and balancing rigorous coursework with early fieldwork—potentially adapting HIPs to boost clinical confidence and reduce attrition in these high-demand fields." The following references are suggested for writing this paragraph:

Afshari A, Khodaveisi M, Sadeghian E. Exploring the educational challenges in emergency medical students: A qualitative study. Journal of Advances in Medical Education & Professionalism. 2021 Apr;9(2):79.

Methods

Why did you use the phenomenological method in this study? Explain.

For enhanced methodological contribution, suggest discussing potential biases in peer mentor checks of reflections and how they were mitigated, which could inform adaptations for health professions training where mentor feedback is common."

Describe in detail how you selected the samples. Which of the purposive methods did you use?

**Do you want your identity to be public for this peer review?** For information about this choice, including consent withdrawal, please see our Privacy Policy

Reviewer #1: No

Reviewer #3: **Yes:** Adeniyi Adeleye

Reviewer #4: No

---

## [Author Response · Author response to Decision Letter 2]

17 Dec 2025

December 10, 2025

Dear Dr. El-Sayed,

We appreciate the additional comments for strengthening this manuscript (PONE-D-24-02966R1). Our responses are included with the label [Response] and in blue text below.

Reviewer 1

All proposed amendments have been integrated into the manuscript, resulting in a revised and enhanced version. With these modifications in place, I believe the article is now ready for publication.

Reviewer 3

Thank you for taking the time and effort to respond to my suggestions. I hope your work contributes to the broader learning and teaching.

[Response]: We thank the reviewers for their comments and insights, which we believe have substantially improved the quality and clarity of our manuscript.

Reviewer 4

Abstract:

The abstract effectively highlights the gap in qualitative research on high-impact practices (HIPs) for second-year students, providing a compelling rationale for the phenomenographic approach. To enhance readability, consider briefly specifying one example of a 'positive outcome' in each dimension to give readers a more vivid preview of the results."

[Response]: Thank you for this suggestion. While we agree that additional examples in the abstract could be beneficial, providing such examples would cause us to exceed the word limit (300 words).

We believe that providing brief examples would also risk oversimplifying the nuanced and contextually situated nature of our qualitative findings. As such, we have designed the abstract to serve as a conceptual roadmap that invites readers to explore this complexity rather than offering a reductive and superficial snapshot. The findings section is where it is best to save the examples so we can faithfully preserve the power of student voice that is central to our study.

Elaborate slightly on how the summer-bridge program's design influenced these outcomes (e.g., activity types), as this could strengthen the link to practical recommendations without exceeding word limits."

[Response]: Thank you for this suggestion. We have revised the paragraph beginning on Line 200 to include more details on program activities and noted that students frequently reflected on these activities in their weekly reflections.

Condense the discussion of first-year comparisons (lines 88-94) slightly to allocate more space for emerging trends in second-year retention, such as post-pandemic effects, to underscore timeliness."

[Response]: Thank you for this suggestion. We have found that the second-year retention rates have been largely similar over the past decade, with slight upward trends post-pandemic. We have noted these statistics in this paragraph (beginning on Line 88) from the National Student Clearinghouse Research Center and American Institutes for Research.

For greater impact, suggest explicitly noting potential overlaps between dimensions (e.g., how social support influences personal efficacy, as hinted in lines 191-192) to invite future intersectional analyses."

[Response]: Thank you for this suggestion. We have added to the second paragraph in the limitations section, where we discuss future directions (beginning on Line 652), to further elaborate on potential overlaps between dimensions and the importance of future intersectional analyses of these dimensions to better understand high-impact practices that support holistic student development.

To extend applicability, I recommend incorporating a brief discussion of how these challenges manifest in nursing and paramedical students, who often grapple with unique educational hurdles like integrating clinical simulations, managing high-stakes ethical dilemmas, and balancing rigorous coursework with early fieldwork—potentially adapting HIPs to boost clinical confidence and reduce attrition in these high-demand fields." The following references are suggested for writing this paragraph: Afshari A, Khodaveisi M, Sadeghian E. Exploring the educational challenges in emergency medical students: A qualitative study. Journal of Advances in Medical Education & Professionalism. 2021 Apr;9(2):79.

[Response]: We appreciate this thoughtful suggestion to increase the impact of our findings. We have added a final sentence in the limitations section acknowledging that future research could examine how high-impact practices support students across different academic and disciplinary contexts, including those with specialized training requirements. We have also included the aforementioned reference. However, given the focus of our study is on second year undergraduate students and the need to maintain a reasonable manuscript length, we felt that a more detailed discussion of discipline-specific challenges in such health professions programs would be beyond the scope of this paper.

Why did you use the phenomenological method in this study? Explain.

[Response]: Thank you for this inquiry about our methodological choice. We have expanded the paragraph beginning on Line 249 to clarify our rationale for using phenomenography. We now explain that phenomenography is particularly suitable for this study because students participating in the same program activities may foreground and interpret different aspects of their experiences, and we clarify how variation theory helps us describe and catalog the range of perceived outcomes from students' engagement in high-impact practices.

For enhanced methodological contribution, suggest discussing potential biases in peer mentor checks of reflections and how they were mitigated, which could inform adaptations for health professions training where mentor feedback is common."

[Response]: Thank you for this suggestion. We have clarified in the paragraph beginning on Line 292 that peer mentor checks were administrative rather than evaluative. Peer mentors checked reflections for completion and verified that students were participation in appropriate activities. However, mentors were not part of the research team and did not provide substantive feedback that would have influenced the content of the reflections, which reduced the potential of biasing or influencing students' self-reported experiences.

Describe in detail how you selected the samples. Which of the purposive methods did you use?

[Response]: Thank you for this question. We have clarified in the paragraph beginning on Line 222 that our sampling approach combined convenience and purposive sampling. We now explicitly identify that we used criterion sampling by selecting a program where students engaged in high-impact practices across four dimensions and submitted structured reflections on these experiences.

We look forward to the review process and to providing any additional information that would be helpful. Thank you very much for your time and thoughtful consideration.

Sincerely,

Stanley M. Lo, Ph.D.

Teaching Professor, Cell and Developmental Biology

Co-Director, Joint Doctoral Program in Mathematics and Science Education

Affiliate Faculty, Research Ethics Program

University of California San Diego

---

## [Editor Report · Decision Letter 2]

21 Dec 2025

In their own words: A qualitative examination of student experiences with high-impact practices during the second-year transition

PONE-D-24-02966R2

Dear Author,

We’re pleased to inform you that your manuscript has been judged scientifically suitable for publication and will be formally accepted for publication once it meets all outstanding technical requirements.

Kind regards,

Ahmed Abdelwahab Ibrahim El-Sayed

Academic Editor

PLOS One

Additional Editor Comments (optional):

Dear Authors,

Your contribution is original. Thank you for all your efforts done to enhance your study. I can accept your manuscript in its current form for publication in PLOS ONE. Congratulations.

---

## [Editor Report · Acceptance letter]

PONE-D-24-02966R2

PLOS One

Dear Dr. Lo,

I'm pleased to inform you that your manuscript has been deemed suitable for publication in PLOS One. Congratulations! Your manuscript is now being handed over to our production team.

Kind regards,

on behalf of

Dr. Ahmed Abdelwahab Ibrahim El-Sayed

Academic Editor

PLOS One